# Effectiveness of virtual reality-based rehabilitation on the upper extremity motor function of stroke patients: A protocol for systematic review and meta-analysis

Jiali Zhang[1], Jie Yang[1]*, Qiuzhu Xu[2], Yan Xiao[1], Liang Zuo[3], Enli Cai[4]*

**1** Kunming Municipal Hospital of Traditional Chinese Medicine, The Third Affiliated Hospital of Yunnan University of Chinese Medicine, Kunming, Yunnan Province, China, **2** Haikou Orthopedic and Diabetes Hospital of Shanghai Sixth People's Hospital, Haikou, Hainan Province, China, **3** The Second Affiliated Hospital of Yunnan University of Traditional Chinese Medicine, Kunming, Yunnan Province, China, **4** Yunnan University of Traditional Chinese Medicine, Kunming, Yunnan Province, China

* y15925112913@163.com (EC); kmszcssd@163.com (JY)

## Abstract

### Introduction

Upper extremity deficits (UED) is a common and impactful complication among stroke survivors. Virtual reality (VR)-based rehabilitation holds potential for enhancing rehabilitation intensity and engagement by stimulating tasks. While several clinical studies have examined the effectiveness and safety of VR-based rehabilitation, there is a need for further research to improve consistency in outcomes.

### Materials and methods

The study will incorporate randomized controlled trials (RCTs) concerning the effects of VR-based rehabilitation on upper extremity (UE) function in stroke survivors. A comprehensive search of databases including PubMed, Embase, Cochrane Library, Web of Science, Scopus, Cinahl, China National Knowledge Infrastructure (CNKI), Wan-fang, and Chinese Biology Medicine Database will be performed from inception to the start of the study. Primary outcomes will focus on upper limb motor function assessments such as the Fugl-Meyer Upper Extremity (FMUE), Box and Block Test (BBT), Wolf Motor Function Test (WMFT), and Action Research Arm Test (ARAT). Secondary outcomes related to activities of daily living will include the Barthel Index (BI) and Functional Independence Measure (FIM). Research selection, data extraction, and quality assessment will be independently conducted by two researchers. The recently revised Cochrane risk of bias tool will be employed to evaluate study quality. Meta-regression and subgroup analyses will be utilized to identify effective therapy delivery modes and patterns. The assessment, development, and evaluation of recommendations approach will be applied to achieve a robust conclusion.

**Data Availability Statement:** No datasets were generated or analysed during the current study. All

relevant data from this study will be made available upon study completion.

**Funding:** The study is supported by a grant from "Research on the Integrated Nursing Intervention Model of Traditional Chinese and Western Medicine for Early Recovery Patients with Ischemic Stroke Based on KTA" (Project No. 202101AZ070001-221) in Yunnan Province, China. No funding bodies had any role in study design, data collection and analysis, decision to publish, or preparation of the manuscript.

**Competing interests:** The authors have declared that no competing interests exist.

## Discussion

This study provides a rigorous synthesis to evaluate optimal parameters—specifically intensity and duration—for VR-based rehabilitation interventions aimed at enhancing UE function in stroke survivors. Our secondary objective is to assess the impact of these parameters on rehabilitation outcomes. We anticipate an accurate, transparent, and standardized review process that will yield evidence-based recommendations for integrating VR technology into treating upper extremity dysfunction in stroke patients, offering clinicians effective strategies to enhance upper limb function.

## Introduction

Stroke is a leading cause of disability worldwide, affecting almost 14 million people annually [1, 2]. While stroke mortality rates show a declining trend, the number of individuals experiencing the consequences of stroke is increasing due to population growth and ageing [3]. This upward trend has led to substantial, enduring disabilities in adults [4].

Approximately 75% of stroke survivors will demonstrate enduring deficits in motor control of their arm and hand, leading to enormous personal and societal consequences [5]. This increase in the number of stroke survivors highlights the growing need for rehabilitation services [6]. A common and severe disabling complication of stroke is UED [7]. These deficits persist partly due to the failure of current nonrepresentational approaches to substantially reduce upper-limb impairment [8]. Common manifestations of UED include loss of strength, reduced flexibility, abnormal cooperative interaction incursion, and muscle tension disorders [9]. These impairments can cause disabilities in common activities such as reaching, picking up objects, and holding onto objects [10].

Moreover, the restoration of UE function is a complex process with poor prognoses [11], which significantly affects patients' independence in daily activities and greatly reduces their quality of life. This places a considerable burden on both families and society. Consequently, it is crucial to enhance the functional capacity of the UE and promote greater levels of independence in individuals after a stroke. Contemporary clinical strategies for UE rehabilitation rely on fostering neuroplasticity post-brain impairments [12]. Intensive and extensive task-specific training emphasizing numerous repetitions, has emerged as crucial in motor therapy following a stroke [13].

According to the guideline, postural training and task-oriented upper limb training have the potential to positively influence on upper limb motor control [14]. Currently, the field of neurorehabilitation encompasses several technologies that hold promise for addressing various neurological dysfunctions [15]. Among these, VR stands out as an innovative intervention in rehabilitation nursing, offering an enriched environment conducive to task-specific training and delivering multimodal feedback to promote functional recovery [16]. VR interventions for motor impairments showed positive rehabilitative effects in stroke survivors [17]. The three fundamental concepts of VR are immersion, imagination, and interaction [18]. Patients can immerse themselves in simulated scenarios, interact with their environment by engaging imagery, and receive real-time feedback, fostering immersive experiences conducive to motor rehabilitation. In parallel with usual rehabilitation therapy programs, VR not only supplements existing strategies but also motivates patients to engage in more purposeful practices, thereby intensifying the effectiveness of targeted movements.

However, there is still no consensus in the field of UE rehabilitation regarding the specific types, duration, and intensity of VR training required to assess its clinical effectiveness [19]. Moreover, a previous meta-analysis indicated a significant increase in the utilization of VR-based training for UE rehabilitation, resulting in varied outcomes. To effectively assess the impact of VR training in improving upper limb dysfunction post-stroke, it is essential to establish a comprehensive and standardized protocol for systematic reviews and meta-analyses. Our primary objective is to develop this standardized protocol to assess the effectiveness of VR-based rehabilitation in enhancing upper limb motor function among adult stroke survivors. Additionally, we aim to investigate the intensity and duration of VR interventions to optimize outcomes related to upper limb functionality. The positive findings from this study may prompt further research into the optimal dosing of VR training, ultimately advancing clinical practice for stroke rehabilitation and contributing to future clinical practice guidelines.

## Materials and methods

For the design and reporting of this systematic review, we will strictly follow the Preferred Reporting Items for Systematic Reviews and Meta-Analysis Protocols (PRISMA-P) 2015 [20] and AMSTAR2 [21]. The PRISMA-P checklist is shown in S1 Table.

### Criteria for study selection

In general, studies will be screened and selected based on PICOS format as follows:

**Types of study.**   Only RCTs published in English from inception to April 1, 2024, will be eligible for inclusion. Studies must include at least two groups: one undergoing VR-based training combined with conventional therapy, and a control group receiving only conventional therapy. Studies evaluating any degree of intensity and different duration within the realm of VR will be incorporated. Additionally, studies employing both immersive and non-immersive VR modalities, as well as those utilizing commercially available gaming consoles, will be eligible for inclusion.

**Types of participants.**   Patients diagnosed with stroke will be considered regardless of their age, sex, severity, or disease duration [22].

**Types of intervention.**   The experimental group (EG) will receive VR-based training in conjunction with conventional therapy.

**Types of control.**   The control group (CG) will be administered only conventional therapy, which includes conventional training, occupational therapy, physical therapy, usual care, or any rehabilitation activities aimed at addressing impairment, activity, or participation levels.

**Types of outcome measure.**   The primary outcomes focused on functionality and its limitations [23]. This included indicators of the ability to perform an UE function (using the arm and hand) and could include tools such as: (1) FMUE [24]; (2) BBT [25]; (3) WMFT [26]; (4) ARAT [27]. Secondary outcomes focused on activities of daily living, such as BI [28] and FIM [29]. Alternatively, additional outcome measures of interest pertaining to the restoration of upper limb functionality following a stroke will be considered.

### Data sources and search strategy

PubMed, Embase, Cochrane Library, Web of Science, Scopus, Cinahl, CNKI, Wan-fang, and Chinese Biology Medicine Database will be comprehensively searched from inception to the start of the study. Medical subject heading (MeSH) terms related and text words will be adopted, mainly including stroke, cerebrovascular disorders, virtual reality, virtual reality

exposure therapy, upper extremity, upper limb, arm. Grey literature such as theses and articles located via the snowball, dissertations and conference proceedings, will also be consulted.

## Data screening and extraction

Two independent authors conducted an initial assessment of studies for potential incorporation. The screening process encompassed the scrutiny of titles and abstracts pinpointed during the search, succeeded by a comprehensive analysis of the complete texts against the predetermined inclusion standards. Excluded studies after full-text appraisal will be meticulously recorded and elucidated concerning the rationale for their exclusion. Any discrepancies in data extraction were resolved through consensus or by consulting a third author. The research flowchart is presented in **Fig 1**.

Data extraction will be performed with a pre-piloted, standardized form. The following information will be extracted: journal title, first author, publication year, country, patient demographics (stroke recovery stage, stroke duration, sample size, sex, mean age), control and experimental parameters (VR session duration, VR training frequency, VR period), outcomes, duration of intervention, among others. Any discrepancies will be resolved through consultation with a third author.

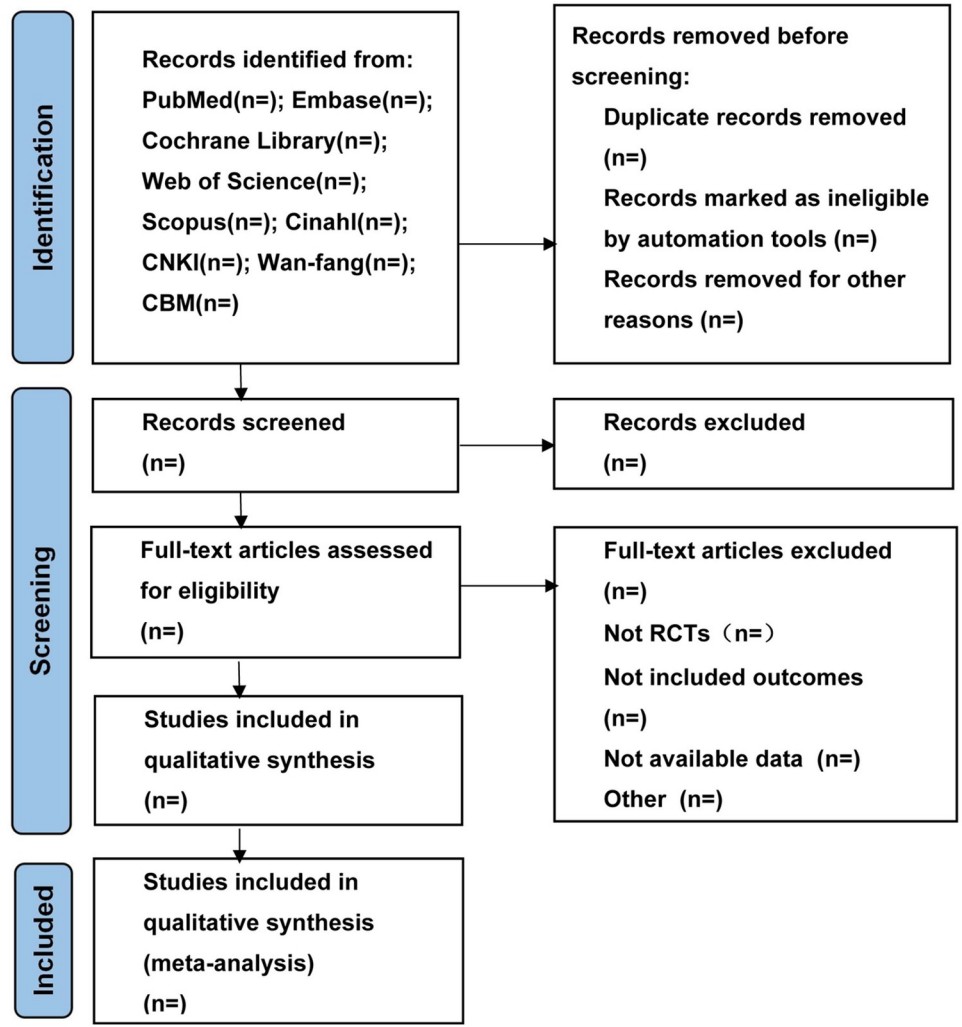

**Fig 1. Flow diagram of study selection process.** (n): is the number of articles that will be included at each stage.

## Data synthesis and analysis

Data synthesis will be conducted utilizing STATA (version 17.0, StataCorp LLC). The outcomes included in the analysis comprised continuous data. Pooled results were estimated by calculating the mean difference (MD) or standardized mean difference (SMD), along with 95% confidence intervals (CI) [30]. The *p*-values were two sided, with an alpha level of 0.05 considered significant [31].

## Assessment of risk of bias

Two reviewers will independently evaluate the risk of bias using the RoB 2 tool (revised tool for risk of bias in randomized studies) [32]. This tool encompasses seven domains: random sequence generation, allocation concealment, blinding of participants and personnel, blinding of outcome assessment, incomplete outcome data, selective reporting, and other biases. The risk of bias will be categorized as "low", "unclear", or "high" for each domain. These seven domains will be appraised independently by two reviewers, and discrepancies will be addressed by consulting a third reviewer.

## Assessment of heterogeneity

In evaluating heterogeneity between trials, the $I^2$ statistic will be employed. The methodology will entail the following procedures: if the $I^2$ test is <50%, the fixed effects model will be applied for data synthesis. Conversely, the random-effects model will be applied for data synthesis if the $I^2$ test is between 50% and 75% [33]. If the $I^2$ test is >75%, meta-regression analyses and subgroup analysis will be conducted to identify possible reasons. A sensitivity analysis will be conducted to evaluate the robustness of the primary decisions made during the review process.

## Assessment of evidence quality

For each aggregated or individual effect size (ES) of functioning, the certainty of the evidence body will be independently rated by two reviewers with GRADE [34]. Any inconsistencies will be resolved through consultation with a third reviewer. Summary of findings tables will be generated utilizing GRADEPro-GDT.

Information on all primary and secondary outcomes derived from our review will be incorporated. The quality of the evidence will be undergo evaluation based on five factors: 1) limitations in trial design and execution of available trials; 2) indirect evidence; 3) unexplained heterogeneity or inconsistency of results; 4) imprecision in effect estimates; and 5) potential publication bias.

If the number of studies exceeds ten, a funnel plot will be constructed and scrutinized to investigate potential biases in studies and publications in accordance with the guidelines outlined in the Cochrane Handbook for Systematic Reviews of Interventions [32]. Begg's and Egger's tests will be employed to evaluate the publication bias of these trials and generate the publication bias plot. If the funnel plots are found to be asymmetrical, we will try to interpret funnel plot asymmetry [35].

## Ethical considerations

This review will not entail the utilization of private individual information nor infringe upon patient rights, thus obviating the necessity for ethical approval. The results of this review will be disseminated through academic publications and conference presentations.

## Discussion

UE function deficits represent a prevalent challenge encountered by stroke survivors, significantly associated with increased levels of physical impairment, disability, and reduced quality of life. Moreover, UE function impairment poses a significant impact on stroke patients, their families and the broader societal framework. The restoration of optimal UE function is crucial for reinstating independence in daily activities.

Over the past twenty years, there has been a notable increase in research on the effectiveness of VR in treating motor function. The use of VR training to enhance arm motor function after a stroke shows great promise. This method allows individuals to participate in engaging training sessions that involve multiple repetitions, prominent stimuli, and challenging tasks, all of which are believed to elicit adaptive neuroplastic changes [36]. The literature supports the notion that VR interventions may yield favorable outcomes on the impairment of UE, potentially outperforming conventional therapeutic approaches. Previous research [37] has illustrated that VR-based training technology has the potential to induce cortical reorganization of neuromotor pathways. Before VR training, activation occurred bilaterally in the patient's primary motor cortex, ipsilateral sensorimotor cortex, and motor accessory area cortex. Following the training, these regions were inhibited, while contralateral sensorimotor cortical areas were activated, thereby promoting compensation and performance of lost motor functions. The use of VR technology in rehabilitation settings allows patients to receive prompt feedback on their task execution, along with visual and auditory stimuli that often capture their interest. VR engenders motivation for active engagement in therapy sessions, as patients derive enjoyment from participating in tasks. Moreover, VR interventions can enhance stability in activities of daily living, thereby augmenting the independence of stroke patient [38].

Currently, the field of UE rehabilitation faces the challenge of customizing VR-based training to meet the distinct requirements of individuals. This poses a potential obstacle for clinicians who wish to integrate VR into clinical practice. Therefore, developing a comprehensive feasibility protocol is essential to amalgamate evidence regarding various types, frequencies, duration, and intensities of VR-based rehabilitation, ensuring clinical effectiveness in addressing UED among stroke survivors.

To the best of our knowledge, this is the first systematic review and meta-analysis to synthesize the effectiveness of VR-based rehabilitation on UED in stroke patients. This paper outlines a rigorous systematic review protocol to evaluate the effectiveness and safety of VR in rehabilitating UE function. This proposed review will be conducted following the latest guidelines of the Cochrane Handbook for Systematic Reviews of Interventions. The statistical method of meta-regression will be used to investigate potential factors affecting VR outcomes. Subsequently, subgroup analyses will be conducted based on the findings. In addition, the outcome indicators formulated for evaluating upper limb function were meticulous and covered a wide array of outcome measures across the diverse RCTs included. We will initially extract dose-effect relationships concerning training volume to provide guidance to clinicians and practitioners in formulating effective VR-based training protocols for UE motor function. Further research is needed to reveal optimal dose-response relationships following VR training.

Furthermore, we will address various relevant considerations in the systematic review and meta-analysis process. While an exhaustive literature search will be conducted, it is acknowledged that not all relevant RCTs studies adhering to the protocol may be included. We will exercise caution in interpreting results, particularly in instances of limited study and patient inclusion, as well as in trials employing multiple treatment methodologies. Another anticipated limitation of this study is the heterogeneity related to variations in stroke severity among participants and differences in VR intervention methodologies. These factors may affect the

generalizability and reliability of our findings. To address these issues, we will perform subgroup analyses based on different VR intervention methods and apply meta-regression techniques, as data allow, to assess the influence of stroke severity and VR characteristics on the outcomes. We will also exercise caution in interpreting our results, especially with limited studies or small sample sizes, and explicitly outline the limitations and uncertainties of our findings.

Despite these potential limitations, our aspiration is for the study outcomes to serve as a valuable synthesis of the available evidence, offering preliminary insights for both current clinical practices addressing UED and guiding future research endeavors. We believe this protocol is promising and it will empower the healthcare professionals, caregivers and especially stroke patients with UED.

## Supporting information

**S1 Table. PRISMA-P (Preferred Reporting Items for Systematic review and Meta-Analysis Protocols) 2015 checklist: Recommended items to address in a systematic review protocol\*.**
(DOC)

## Acknowledgments

The authors thank teacher Cheng Qing for consultation regarding the statistical analysis and study design.

## Author Contributions

**Conceptualization:** Jiali Zhang, Enli Cai.

**Data curation:** Qiuzhu Xu, Yan Xiao, Liang Zuo.

**Formal analysis:** Jiali Zhang, Jie Yang.

**Funding acquisition:** Enli Cai.

**Investigation:** Qiuzhu Xu, Yan Xiao.

**Methodology:** Jiali Zhang, Jie Yang.

**Project administration:** Jiali Zhang.

**Resources:** Qiuzhu Xu, Yan Xiao.

**Software:** Jie Yang, Liang Zuo.

**Supervision:** Enli Cai.

**Validation:** Jiali Zhang, Qiuzhu Xu, Yan Xiao, Liang Zuo.

**Visualization:** Jie Yang.

**Writing – original draft:** Jiali Zhang, Jie Yang.

**Writing – review & editing:** Jiali Zhang, Jie Yang, Enli Cai.

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
