## [Decision Letter · Decision Letter 0]

9 Jul 2024

PONE-D-24-16134Efficacy of virtual reality-based rehabilitation on the upper extremity deficits of stroke patients: A protocol for systematic review and meta-analysisPLOS ONE

Dear Dr. Zhang,

Thank you for submitting your manuscript to PLOS ONE. After careful consideration, we feel that it has merit but does not fully meet PLOS ONE’s publication criteria as it currently stands. Therefore, we invite you to submit a revised version of the manuscript that addresses the points raised during the review process. 

We look forward to receiving your revised manuscript.

Kind regards,

Mohammed Usman Ali

Academic Editor

PLOS ONE

Reviewers' comments:

Reviewer's Responses to Questions

**Comments to the Author**

1. Does the manuscript provide a valid rationale for the proposed study, with clearly identified and justified research questions?

Reviewer #1: Yes

Reviewer #2: Partly

Reviewer #3: Yes

2. Is the protocol technically sound and planned in a manner that will lead to a meaningful outcome and allow testing the stated hypotheses?

Reviewer #1: Yes

Reviewer #2: Partly

Reviewer #3: Yes

3. Is the methodology feasible and described in sufficient detail to allow the work to be replicable?

Reviewer #1: Yes

Reviewer #2: Yes

Reviewer #3: Yes

4. Have the authors described where all data underlying the findings will be made available when the study is complete?

Reviewer #1: Yes

Reviewer #2: Yes

Reviewer #3: Yes

5. Is the manuscript presented in an intelligible fashion and written in standard English?

Reviewer #1: Yes

Reviewer #2: Yes

Reviewer #3: Yes

6. Review Comments to the Author

You may also provide optional suggestions and comments to authors that they might find helpful in planning their study.

Reviewer #1: The manuscript provides a strong rationale for the proposed study, focusing on the efficacy of virtual reality (VR)-based rehabilitation for upper extremity deficits in stroke patients. It addresses a significant and common issue faced by stroke survivors, contributing to the field of neurorehabilitation. The introduction effectively highlights the prevalence and impact of upper extremity deficits post-stroke, and the potential benefits of VR-based rehabilitation. The research question is clearly identified, aiming to synthesize current evidence on the effectiveness of VR interventions, which is a valid academic problem and could significantly enhance clinical practice and rehabilitation strategies.

The protocol is technically sound and planned to lead to meaningful outcomes. The manuscript outlines a systematic approach incorporating randomized controlled trials (RCTs) to assess the effects of VR-based rehabilitation on upper extremity function. The use of comprehensive databases and predefined outcome measures (such as FMUE, BBT, WMFT, and ARAT for primary outcomes, and BI and FIM for secondary outcomes) ensures a structured data collection process. The application of the Cochrane risk of bias tool and planned meta-regression and subgroup analyses add robustness to the methodology. Statistical power analysis is implicit in the extensive data sources and selection criteria, although explicit power analysis details are not provided.

The methodology is feasible and described in sufficient detail to allow replication. The manuscript provides a clear description of the search strategy, inclusion criteria, data extraction, and analysis procedures. The use of independent researchers for study selection and quality assessment ensures objectivity and reproducibility. The detailed explanation of the planned statistical methods, including heterogeneity assessment and evidence quality evaluation, supports the feasibility and robustness of the study design. However, including explicit sample size calculations and more detailed statistical power analysis would strengthen the protocol.

The authors have stated that no datasets were generated or analyzed during the current study and that relevant data will be made available upon study completion. This meets the basic requirement for data availability, but more explicit details on how and where the data will be shared could enhance transparency.

The manuscript is presented in an intelligible fashion and written in standard English. The structure follows a logical flow, and the language is clear and professional. The inclusion of a PRISMA-P checklist ensures adherence to reporting standards, which enhances the clarity and comprehensiveness of the protocol.

The authors might consider addressing potential limitations related to heterogeneity in stroke severity among participants and variations in VR interventions. These factors could impact the generalizability of the findings.

There are no evident ethical concerns regarding the research or publication process. The authors have declared no competing interests and no specific funding, which supports the objectivity of the study.

Overall, the manuscript presents a well-justified, methodologically sound, and feasible protocol for a systematic review and meta-analysis on the efficacy of VR-based rehabilitation for upper extremity deficits in stroke patients. It is suitable for publication.

Reviewer #2: The presented study is a very interesting topic, but there are already two systematic reviews, one from 2018 and another from 2022, with similar maths. In this sense, it is suggested that the authors could reformulate the objective of the study, since the secondary objective will be to understand what is the intensity and duration of the VR intervention to achieve better results in upper limb functionality.

Title should be reformulated to understand the outcome and if it is a systematic review, it is not efficacy but effectiveness what authors want to measure.

The eligibility criteria should be better described (e.g. studies language; search time).

As a protocol of a systematic review with metanalysis submission register could be described.

In PICOS study design the C is not described, what will be compared?

The structure of the abstract and the paper should be similar and following PLOS one submission guidelines. If authors present discussion it should be present in both parts. In this case we have conclusion in the abstract and discussion on the paper. Also it should be Materials and Methods and not Methods and Analysis or just Methods as presented.

Vancouver rules should be followed, reference number is not in a rectangular parentheses.

Reviewer #3: Under the sub topic "Assessment of heterogenicity"

Please correct the tense of the sentences to future tense (from the lines 149-153)

7. PLOS authors have the option to publish the peer review history of their article (what does this mean?). If published, this will include your full peer review and any attached files.

Reviewer #1: **Yes: **Ana Isabel Correia Matos de Ferreira Vieira

Reviewer #2: No

Reviewer #3: **Yes: **Surangika Wadugodapitiya

---

## [Author Response · Author response to Decision Letter 0]

19 Aug 2024

Dear Prof. Editors,

I wish to re-submit the manuscript titled "Effectiveness of virtual reality-based rehabilitation on the upper extremity motor function of stroke patients: A protocol for systematic review and meta-analysis". The manuscript ID is PONE-D-24-16134. 

We sincerely appreciate the thoughtful suggestions and insights provided by you and the reviewers. Your valuable feedback has been instrumental in guiding our revisions, and we believe it has significantly improved the quality of our manuscript. We look forward to collaborating with you and the reviewers to bring this manuscript closer to publication in PLOS ONE.

The manuscript has been thoroughly reviewed, and all necessary changes have been made in accordance with the reviewers' suggestions. The revised sections are highlighted in red for your convenience.

I also wish to apologize for an oversight in the first author’s affiliations. The manuscript currently lists "Kunming Municipal Hospital of Traditional Chinese Medicine", but I inadvertently omitted "The Third Affiliated Hospital of Yunnan University of Chinese Medicine". Per institutional regulations, both affiliations should be included. We kindly request your permission to add this additional affiliation.

We have strived to improve the manuscript while keeping the content and framework intact. Below, you will find our point-by-point responses to the comments raised by you and reviewers. All modifications are clearly marked in red in the revised document for your convenience.

Thank you for your consideration, and I look forward to your response.

Sincerely, 

Jiali Zhang 

The following is a point-to-point response to you.

Point 1: Laboratory protocols

Response: Our meta-protocol primarily serves as a methodological framework for systematic reviews and meta-analyses, rather than a laboratory-specific experimental procedure. Thus, it may not be ideally suited for publication on protocols.io. Our framework focuses on evidence synthesis across multiple studies, which may not align with the format of protocols.io.

Point 2: Manuscript formatting 

Response: We have made the necessary revisions to ensure that our manuscript complies with the PLOS ONE formatting guidelines. 

Please check the changes highlighted in red on page 11, lines 261 to 271, and page 12, lines 272 to 277.

Point 3: Data availability

Response: The data availability statement has been updated to: "Data availability: All relevant data are within the manuscript and its Supporting Information files."

Please check the updated statement located on page 1, line 23, which is highlighted in red.

The following is a point-to-point response to the reviewers.

Reviewer #1: 

1.However, including explicit sample size calculations and more detailed statistical power analysis would strengthen the protocol.

Response: We would like to respectfully clarify that our article is not a meta-analysis, but a methodological framework to guide future meta-analysis in a specific field. Consequently, traditional sample size calculations typically used in primary research studies, are not applicable in this particular context. 

Moreover, we wish to emphasize that our statistical methods do not include power analysis, as it is not relevant to our research design. As a general rule, a meta-analysis can be statistically synthesized with more than 3 articles included. Please refer to the 'Data Synthesis and Analysis' section for details.

2. The authors have stated that no datasets were generated or analyzed during the current study and that relevant data will be made available upon study completion. This meets the basic requirement for data availability, but more explicit details on how and where the data will be shared could enhance transparency.

Response: The data availability statement has been updated to: "Data availability: All relevant data are within the manuscript and its Supporting Information files."

Please check the updated statement located on page 1, line 23, which is highlighted in red.

3.The authors might consider addressing potential limitations related to heterogeneity in stroke severity among participants and variations in VR interventions. These factors could impact the generalizability of the findings.

Response: To address the potential limitations related to heterogeneity and VR interventions, we have enhanced our manuscript as follows:

(i)Subgroup Analyses: We will stratify trials based on different VR intervention methodologies to investigate the effects of diverse approaches. 

(ii)Meta-Regression: Depending on data availability, we will utilize meta-regression to assess the influence of stroke severity and intervention characteristics.

(iii)Result Interpretation: We will interpret the results with caution, particularly in cases of limited studies or small sample sizes.

Please check the revisions highlighted in red on line 246 of page 10 and lines 247-254 of page 11.

Reviewer #2: 

1.The presented study is a very interesting topic, but there are already two systematic reviews, one from 2018 and another from 2022, with similar maths. In this sense, it is suggested that the authors could reformulate the objective of the study, since the secondary objective will be to understand what is the intensity and duration of the VR intervention to achieve better results in upper limb functionality.

Response: We aim to explore the intensity and duration of VR interventions while maintaining our primary goal of developing a standardized protocol to evaluate the effectiveness of VR rehabilitation training in enhancing upper limb motor function among adult stroke survivors in the present study. We have revised the manuscript to incorporate these changes, as reflected in the modified sections below:

"Discussion

This study provides a rigorous synthesis to evaluate optimal parameters—specifically intensity and duration—for VR-based rehabilitation interventions aimed at enhancing UE function in stroke survivors. Our secondary objective is to assess the impact of these parameters on rehabilitation outcomes. We anticipate an accurate, transparent, and standardized review process that will yield evidence-based recommendations for integrating VR technology into treating upper extremity dysfunction in stroke patients, offering clinicians effective strategies to enhance upper limb function."

"To effectively assess the impact of VR training in improving upper limb dysfunction post-stroke, it is essential to establish a comprehensive and standardized protocol for systematic reviews and meta-analyses. Our primary objective is to develop this standardized protocol to assess the effectiveness of VR-based rehabilitation in enhancing upper limb motor function among adult stroke survivors. Additionally, we aim to investigate the intensity and duration of VR interventions to optimize outcomes related to upper limb functionality. The positive findings from this study may prompt further research into the optimal dosing of VR training, ultimately advancing clinical practice for stroke rehabilitation and contributing to future clinical practice guidelines."

Please check the updated sections on Page 2 (line 48), Page 3 (lines 49-54), Page 4 (lines 96-98), and Page 5 (lines 99-105), with revised content highlighted in red for your convenience.

2.Title should be reformulated to understand the outcome and if it is a systematic review, it is not efficacy but effectiveness what authors want to measure.

Response: The title has been amended to "Effectiveness of virtual reality-based rehabilitation on the upper extremity motor function of stroke patients: A protocol for systematic review and meta-analysis" (page 1, line 1). Additionally, "efficacy" has been replaced with "effectiveness" throughout the manuscript (pages 2, 4, 5, 9, and 10), with modifications highlighted in red.

3.The eligibility criteria should be better described (e.g. studies language; search time).

Response: Eligibility criteria now specify RCTs published in English from inception to April 1, 2024, requiring at least two groups: one undergoing VR-based training plus conventional therapy, and a control group with only conventional therapy.

Please check the update marked in red on page 5, lines 113-116.

4.As a protocol of a systematic review with metanalysis submission register could be described.

Response: Our systematic review protocol has been registered with PROSPERO (registration number: CRD42024574631). Please check the updates marked in red on page 3, line 56.

5. In PICOS study design the C is not described, what will be compared?

Response: We have clarified the control group (C) in the PICOS framework by introducing a separate "Types of control" section. 

The updated content is marked in red on page 5, line 122 and page 6, lines 123-127 of the revised manuscript as shown below:

Types of intervention. The experimental group (EG) will receive VR-based training in conjunction with conventional therapy.

Types of control. The control group (CG) will be administered only conventional therapy, which includes conventional training, occupational therapy, physical therapy, usual care, or any rehabilitation activities aimed at addressing impairment, activity, or participation levels.

6. The structure of the abstract and the paper should be similar and following PLOS one submission guidelines. If authors present discussion it should be present in both parts. In this case we have conclusion in the abstract and discussion on the paper. Also it should be Materials and Methods and not Methods and Analysis or just Methods as presented.

Response: We have revised the manuscript to comply with PLOS ONE submission guidelines by changing the "Conclusion" section in the abstract to "Discussion" (page 2, line 47), with changes marked in red. This adjustment better integrates relevant content, as our article focuses on the design and rationale of a systematic review and meta-analysis protocol rather than presenting direct findings. Since the research is ongoing, a separate conclusion is unnecessary at this stage.

We have also revised the titles from "Methods" and "Methods and Analysis" to "Materials and Methods" (page 2, line 31; page 5, line 107), with changes marked in red.

7. Vancouver rules should be followed, reference number is not in a rectangular parentheses.

Response: We have reviewed the manuscript for adherence to the PLOS ONE Formatting Guidelines. We would like to clarify that, according to the Vancouver referencing style, reference numbers are presented within rectangular parentheses, which is consistently applied throughout the manuscript. We appreciate your careful consideration of our work.

Reviewer #3: 

1.Under the sub topic "Assessment of heterogenicity"

Please correct the tense of the sentences to future tense (from the lines 149-153)

Response: We regret the previous errors and have corrected them accordingly. The updated paragraph is as follows:

"In evaluating heterogeneity between trials, the I² statistic will be employed. The methodology will entail the following procedures: if the I² test is <50%, the fixed effects model will be applied for data synthesis. Conversely, the random-effects model will be applied for data synthesis if the I² test is between 50% and 75% [1]."

Please check the red-highlighted changes on page 7, line 171, and page 8, lines 172-174.

References

[1] Wong SH, Gao QY, Tsoi KKF, Wu WKK, Tam L-s, Lee N, et al. Effect of immunosuppressive therapy on interferon γ release assay for latent tuberculosis screening in patients with autoimmune diseases: a systematic review and meta-analysis. Thorax. 2016;71(1):64-72. doi: 10.1136/thoraxjnl-2015-207811

---

## [Decision Letter · Decision Letter 1]

22 Oct 2024

Effectiveness of virtual reality-based rehabilitation on the upper extremity motor function of stroke patients: A protocol for systematic review and meta-analysis

PONE-D-24-16134R1

Dear Dr. Zhang,

We’re pleased to inform you that your manuscript has been judged scientifically suitable for publication and will be formally accepted for publication once it meets all outstanding technical requirements.

Kind regards,

Domiziano Tarantino, MD

Academic Editor

PLOS ONE

Additional Editor Comments (optional):

Reviewers' comments:

Reviewer's Responses to Questions

**Comments to the Author**

1. Does the manuscript provide a valid rationale for the proposed study, with clearly identified and justified research questions?

Reviewer #2: Yes

Reviewer #3: Yes

2. Is the protocol technically sound and planned in a manner that will lead to a meaningful outcome and allow testing the stated hypotheses?

Reviewer #2: Yes

Reviewer #3: Yes

3. Is the methodology feasible and described in sufficient detail to allow the work to be replicable?

Reviewer #2: Yes

Reviewer #3: Yes

4. Have the authors described where all data underlying the findings will be made available when the study is complete?

Reviewer #2: Yes

Reviewer #3: Yes

5. Is the manuscript presented in an intelligible fashion and written in standard English?

Reviewer #2: Yes

Reviewer #3: Yes

6. Review Comments to the Author

You may also provide optional suggestions and comments to authors that they might find helpful in planning their study.

Reviewer #2: After carefully reviewing the revised version of the manuscript, I am satisfied that the authors have adequately addressed the concerns raised in the initial review. The revisions have improved the clarity and overall quality of the paper, and I now consider it to be in a suitable condition for publication. I have no further suggestions or comments.

Reviewer #3: The author answered the comments which I have raised. I accept the revised version of the manuscript

7. PLOS authors have the option to publish the peer review history of their article (what does this mean?). If published, this will include your full peer review and any attached files.

Reviewer #2: No

Reviewer #3: **Yes: **Surangika Wadugodapitiya

---

## [Editor Report · Acceptance letter]

29 Oct 2024

PONE-D-24-16134R1 

PLOS ONE

Dear Dr. Zhang, 

I'm pleased to inform you that your manuscript has been deemed suitable for publication in PLOS ONE. Congratulations! Your manuscript is now being handed over to our production team.

Kind regards, 

on behalf of

Dr. Domiziano Tarantino 

Academic Editor

PLOS ONE